# Clinical Utility and Limitation of Diagnostic Ability for Different Degrees of Dysplasia of Intraductal Papillary Mucinous Neoplasms of the Pancreas Using ^18^F-Fluorodeoxyglucose-Positron Emission Tomography/Computed Tomography

**DOI:** 10.3390/cancers13184633

**Published:** 2021-09-15

**Authors:** Yuto Hozaka, Hiroshi Kurahara, Hideyuki Oi, Tetsuya Idichi, Yoichi Yamasaki, Yota Kawasaki, Kiyonori Tanoue, Megumi Jinguji, Masatoyo Nakajo, Atsushi Tani, Akihiro Nakajo, Yuko Mataki, Yoshihiko Fukukura, Hirotsugu Noguchi, Michiyo Higashi, Takashi Yoshiura, Akihide Tanimoto, Takao Ohtsuka

**Affiliations:** 1Department of Digestive Surgery, Breast and Thyroid Surgery, Graduate School of Medical and Dental Sciences, Kagoshima University, 8-35-1 Sakuragaoka, Kagoshima City 890-8520, Japan; k6958371@kadai.jp (Y.H.); h-krhr@m3.kufm.kagoshima-u.ac.jp (H.K.); k6207619@kadai.jp (H.O.); k3352693@kadai.jp (T.I.); k3825645@kadai.jp (Y.Y.); k5968102@kadai.jp (Y.K.); wilson@m.kufm.kagoshima-u.ac.jp (K.T.); anakajo@m.kufm.kagoshima-u.ac.jp (A.N.); mataki@m.kufm.kagoshima-u.ac.jp (Y.M.); 2Department of Radiology, Graduate School of Medical and Dental Sciences, Kagoshima University, 8-35-1 Sakuragaoka, Kagoshima City 890-8520, Japan; megu@m.kufm.kagoshima-u.ac.jp (M.J.); masatoyo@m3.kufm.kagoshima-u.ac.jp (M.N.); at-tani@m.kufm.kagoshima-u.ac.jp (A.T.); fukukura@m.kufm.kagoshima-u.ac.jp (Y.F.); yoshiura@m3.kufm.kagoshima-u.ac.jp (T.Y.); 3Department of Pathology, Graduate School of Medical and Dental Sciences, Kagoshima University, 8-35-1 Sakuragaoka, Kagoshima City 890-8520, Japan; h-noguchi@kufm.kagoshima-u.ac.jp (H.N.); east@m2.kufm.kagoshima-u.ac.jp (M.H.); akit09@m3.kufm.kagoshima-u.ac.jp (A.T.)

**Keywords:** IPMN, FDG, PET, mural nodule, high-grade dysplasia, glucose transporter 1, GLUT-1

## Abstract

**Simple Summary:**

Elucidating risk factors for different degrees of dysplasia of intraductal papillary mucinous neoplasms (IPMNs) of the pancreas is important in determining strategies for management. In this study, we searched for risk factors for different degrees of dysplasia of IPMNs. Our study indicated that there were no useful factors that significantly differentiated low-grade dysplasia and high-grade dysplasia; however, ^18^F-fluorodeoxyglucose–positron emission tomography/computed tomography is useful for differentiating between non-invasive and invasive IPMNs. Our results offer critical information that may determine surgical treatment strategies.

**Abstract:**

The diagnostic value of ^18^F-fluorodeoxyglucose (FDG) uptake in the management of intraductal papillary mucinous neoplasms (IPMNs) of the pancreas remains unclear. This study aimed to assess the role of FDG uptake in the diagnosis of different degrees of dysplasia of IPMNs. We retrospectively analyzed the following three points in 84 patients with IPMNs: (1) risk factors to predict high-grade dysplasia (HGD) and invasive carcinoma (INV); (2) the relationship between FDG uptake and glucose transporter 1 (GLUT-1) expression; and (3) the relationship between FDG uptake and the presence of mural nodules. The histopathological diagnosis was low-grade dysplasia (LGD) in 43 patients, HGD in 16, and INV in 25. The maximum standardized uptake value (SUV-max) was significantly higher in INV than in LGD/HGD (*p* < 0.0001, *p* = 0.0136). The sensitivity and specificity to discriminate INV from LGD/HGD were 80.0% and 86.2%, respectively, using the receiver operator characteristic curve, when the optimal cutoff score of SUV-max was set at 4.03. Those values were not different between HGD and LGD. More than half of HGD patients had low GLUT-1 expression. Taken together, FDG-PET/CT is useful in distinguishing between non-invasive and invasive IPMN. Our results offer critical information that may determine surgical treatment strategies.

## 1. Introduction

Intraductal papillary mucinous neoplasms (IPMNs) of the pancreas originate from the pancreatic duct epithelium, and are characterized by papillary growth, pancreatic duct dilation, and mucin hypersecretion [1,2]. It has been well established that IPMNs represent one of the premalignant lesions of pancreatic carcinoma [1,3]. IPMNs have widespread pathogenicity, ranging from low-grade dysplasia (LGD) to high-grade dysplasia (HGD) and invasive carcinoma (INV), with poor prognosis, similar to pancreatic ductal adenocarcinoma (PDAC) [3,4]. Regarding treatment, IPMNs should be resected oncologically according to the stage of premalignant lesions; however, pancreatectomy has a high perioperative morbidity rate, even at high-volume centers and, therefore, resection is not recommended for all cases [5,6,7,8,9,10]. Malignancy diagnosis plays an important role in the decision of follow-up or resection. The indicators for malignant IPMNs of the pancreas were recommended in the international consensus guidelines (ICGs) known as the Sendai Guidelines (GLs), published by the International Association of Pancreatology in 2006 [4]. The disadvantages of the Sendai criteria include a large number of benign, branched-type IPMNs (BD-IPMN) in the surgical indications, resulting in a lower positive predictive value (PPV) for malignancy. Thus, the revised ICGs in 2012 (known as the Fukuoka GLs) incorporated a clinical management algorithm for BD-IPMN based on two grouped risk factors for malignant IPMN: “high-risk stigmata (HRS)”, and “worrisome feature (WF)”. With this revision, management of BD-IPMN changed from an aggressive to a conservative approach [11]. In clinical practice, this management algorithm using HRS and WF is useful not only for BD-IPMN, but also for both the main duct-type IPMN (MD-IPMN) and the mixed-type IPMN. In addition, the ICGs were revised in 2017, with some minor changes to HRS and WF [12]. However, these indicators inherit the limitation of a low PPV, requiring further improvement. Additionally, we occasionally need more information about the degree of dysplasia of the IPMN, depending on the patient’s age, general condition, tumor location, etc. In particular, during surgical decision making, it is important to determine whether the lesion is a non-invasive or invasive IPMN.

From the perspective of malignancy diagnosis, ^18^F-fluorodeoxyglucose (FDG)-positron emission tomography/computed tomography (PET/CT) is widely performed for diagnosis, staging, and therapeutic effect in various carcinomas [13,14,15,16,17]. FDG-PET is a modality that evaluates cell activity by observing FDG uptake via glucose metabolism. Neoplastic cells with activated glucose uptake and FDG uptake also increase in numbers and, therefore, can be used for evaluation [18]. In previous studies, it has been reported that FDG-PET/CT has excellent malignancy diagnostic ability in IPMN [19,20,21]. However, few previous studies have analyzed results according to the differing degrees of dysplasia [22,23,24,25]. In addition, it remains unclear how the abnormal accumulation of FDG is related to mural nodules presumed to contain an abundance of neoplastic cells. Therefore, in the present study, we investigated (1) the diagnostic ability of FDG-PET/CT, which separated IPMNs associated with INV and HGD, respectively; (2) the relationship between FDG uptake and the expression of glucose transporter 1 (GLUT-1), which is one of the glucose transporters involved in the glucose metabolism of IPMNs; and (3) the relationship between FDG uptake and IPMNs with mural nodules.

## 2. Materials and Methods

### 2.1. Study Design and Patient Population

A total of 100 patients underwent pancreatic resection for IPMN at the Department of Digestive Surgery, Breast and Thyroid Surgery, Kagoshima University, between 2006 and March 2021. After excluding patients (1) who did not undergo preoperative FDG-PET, (2) who underwent recombined section of concomitant PDAC distant from IPMN, and (3) with high blood glucose (>150 mg/dL) before FDG injection, 84 patients were enrolled (Figure 1). We retrospectively reviewed the preoperative imaging data, including computed tomography (CT), magnetic resonance imaging (MRI), endoscopic ultrasonography (EUS), FDG-PET/CT, laboratory data, and clinicopathological data of the patients. For clinicopathological data, we evaluated age, sex, symptoms, tumor location, morphological subtype of IPMN, cystic size, mural nodule height, and histopathological diagnosis. Medical records and imaging data for patients who underwent surgery prior to 2017 were also retrospectively evaluated in accordance with the revised Fukuoka GLs. This study was approved by the Ethics Review Committee of Kagoshima University (approval no. 200277, date of approval: 19 March 2021). Written informed consent was obtained from all enrolled patients.

### 2.2. Surgical Indications

Surgical resection was determined according to the criteria of the GLs of each period, according to the Sendai, Fukuoka, and revised Fukuoka GLs.

### 2.3. Imaging Modalities Protocol

We performed contrast-enhanced CT, followed by MRI and EUS, in all patients during the initial assessment of IPMNs. Then, PET/CT was conducted in patients who were intended to undergo surgical resection. All patients underwent whole-body FDG-PET before treatment using Discovery STE (GE Medical Systems, Milwaukee, WI, USA) devices. Patients fasted for 5 h before receiving an intravenous administration of FDG, and FDG-PET images were acquired 120 min after the administration of FDG. We evaluated the IPMN area with maximum FDG uptake, using contrast-enhanced CT, MRI, and EUS findings as a guide. SUV was automatically calculated as the activity concentration (^18^F-FDG uptake divided by the injected dose of ^18^F-FDG (dose/g body weight)). SUV-max equals the organic radioactivity ((MB q/g)/^18^F-FDG (MBq/g body weight)), and was calculated in the IPMN region with the highest FDG uptake. For example, the value was highest in the mural nodule in some IPMNs, but in the cyst wall or septa in others. The FDG uptake defined positives as SUV levels of 2.0 or higher. Other imaging modalities—including contrast-enhanced CT, MRI, and EUS—were performed during the three months before surgery.

### 2.4. Evaluation of Mural Nodule Height

We measured mural nodule height using contrast-enhanced CT.

### 2.5. Histopathological Diagnosis

According to the World Health Organization classification revised in 2019, the specimens were classified as LGD; HGD, including carcinoma in situ; or INV based on the retrospective pathological findings. LGD was defined as benign, whereas HGD and INV were defined as malignant. The histopathological diagnoses were confirmed by at least two pathologists.

### 2.6. Immunohistochemistry (IHC) and Evaluation of IHC

Resected specimens from 35 IPMN patients (comprising 15 HGD, 10 LGD, and 10 INV patients) were used for IHC analysis. Briefly, paraffin-embedded sections, including the area of the tumor with the highest degree of dysplasia, were sliced at a thickness of 3 μm. After deparaffinization and dehydration, the sections were heated at 121 °C for 10 min for antigen retrieval. Sections were soaked in phosphate-buffered saline (PBS) prior to IHC analysis. The sections were also soaked in 0.3% H_2_O_2_ for 10 min to block endogenous tissue peroxidase, which was followed by treatment with bovine serum for 30 min to reduce nonspecific binding. The sections were incubated with primary mouse monoclonal anti-GLUT-1 (dilution, 1:100; Cat. No sc-377228; Santa Cruz, TX, USA) overnight at 4 °C. Sections were rinsed in PBS and visualized via standard techniques for labeled avidin–biotin immunoperoxidase staining. GLUT-1 was subsequently visualized using a 3,3′-diaminobenzidine (DAB) substrate kit. The slides were counterstained with hematoxylin. Red blood cells were used as positive controls for the intensity of GLUT-1 expression. The method of evaluating IHC for GLUT-1 expression was the same as in the previous study [26]. GLUT-1-positive expression was defined as detectable immunoreaction in cell membrane regions of >10 % of the atypical cells.

### 2.7. Statistical Analysis

Values are expressed as median and range. Differences between the two groups were analyzed using Mann–Whitney U tests for continuous variables. Multiple comparisons were performed using the Steel–Dwass tests. Fisher’s exact probability tests were performed for categorized variables. The relationships between possible predictive factors and malignancy of IPMN were determined using a multivariate logistic regression model. The optimal cutoff score for discriminating differences between two groups was determined by constructing receiver operating characteristic (ROC) curves, based on the sensitivities and specificities at several predetermined cutoff points. Correlations were identified using Spearman’s correlation coefficients. Any *p*-values of <0.05 were considered statistically significant. All statistical analyses were performed using JMP Pro 15 (SAS Institute Inc., Cary, NC, USA) and GraphPad Prism 7 (GraphPad Software, La Jolla, CA, USA).

## 3. Results

### 3.1. Clinicopathological Characteristics of the Patients with IPMN

The patients’ background characteristics are summarized in Table 1. IPMN types were MD-IPMN (*n* = 9; 10.7%), BD-IPMN (*n* = 33; 39.3%), and mixed-IPMN (*n* = 42; 50.0%). The locations were the pancreatic head (*n* = 49; 58.3%), distal pancreas (pancreatic body and tail) (*n* = 23; 27.4%), and multifocal (*n* = 12; 14.3%). The pathological diagnosis was LGD (*n* = 43; 51.2%), HGD (*n* = 16; 19.0%), or INV (*n* = 25; 29.8%). No patient had a mucous mass that was initially judged as a mural nodule.

The comparison between the benign and malignant groups revealed that the following clinicopathological findings were significantly more frequent in patients in malignant group compared with the benign group: male (*p* = 0.0128), presence of symptoms (*p* = 0.0006), high levels of carcinoembryonic antigen (CEA) (*p* = 0.0438), high levels of carbohydrate antigen 19-9 (CA19-9) (*p* = 0.0074), high levels of SUV-max (*p* < 0.001), and positivity for HRS (one or more). Similarly, the main pancreatic duct (MPD) size (*p* = 0.0484) and enhancing mural nodule height (*p* < 0.001) were significantly greater in the malignant group compared with those in the benign group (Table 2). In contrast, there were no significant differences between the two groups in terms of age distribution, coexistence of diabetes mellitus, concomitant pancreatitis, family history of pancreatic carcinoma, high levels of pancreatic amylase (P-AMY), morphological subtype, cystic size, cyst growth rate (≥5 mm/2 years), abrupt change in caliber of pancreatic duct with distal pancreatic atrophy, thickened/enhancing cyst walls, lymphadenopathy, or positivity for WF(s) (one or more).

Next, we analyzed the clinicopathological and imaging findings of the HGD, INV, and LGD groups. The rate of symptoms, CA19-9 levels, and P-AMY levels were significantly higher in the INV group than in the LGD group, but not in the HGD group. Enhancing mural nodule height and SUV-max were significantly higher in the INV group than in the LGD and HGD groups (Table 3). In contrast, when the MPD size was ≥10 mm, a significant difference was observed only in the HGD group, and no difference in the INV group. Representative MRI, EUS, and PET/CT images for IPMN with LGD, HGD, and INV are shown in Appendix A.

### 3.2. Malignancy Predictive Ability of ^18^F-Fluorodeoxyglucose-PET/CT and Mural Nodule Height for Malignant IPMN

The SUV-max in malignant IPMNs was significantly higher than that in the LGD group (*p* < 0.0001) (Figure 2A). Among malignant IPMNs, high SUV-max was significantly more frequent in INV patients than in HGD patients (*p* = 0.0136), but there was no difference in SUV-max between the LGD and HGD groups (*p* = 0.4921) (Figure 2B).

The ROC curve for comparison between malignant IPMN (HGD + INV) and LGD, with a cutoff score of 4.0, is shown in Figure 3A; the area under the curve (AUC) was 0.7714. Sensitivity and specificity were 90.7% and 61.0%, respectively, when FDG uptake was used as a marker for benign/malignant differentiation of IPMNs. Among these 12 malignant IPMNs, IPMNs with HGD accounted for 9 cases (Appendix A). From a different perspective, 9 out of 16 HGD patients (56.3%) did not show FDG uptake. If the ROC curve was subtracted from the discriminability of FDG-PET/CT with INV only from LGD and HGD, the optimal cutoff score was 4.03, AUC was 0.8634, sensitivity was 80.0%, and specificity was 86.2% (Figure 3B).

When the ROC curve was constructed using the mural nodule height measured by CT, MRI, and EUS for benign/malignant discrimination, the optimal cutoff values and AUC were as follows: CT 4.8 mm (AUC 0.762), MRI 5.4 mm (AUC 0.731), and EUS 9.9 mm, (AUC 0.785).

### 3.3. Relationship between ^18^F-Fluorodeoxyglucose Uptake and Glucose Transporter 1 (GLUT-1) Expression in Immunohistochemistry for IPMNs

The selected samples comprised the five patients with the highest SUV-max values and the five patients with the lowest SUV-max values. The representative results of immunohistochemical staining for GLUT-1 are shown in Figure 4. In the LGD group, the incidence of GLUT-1-positive expression in the membrane was 20% (2/10) (Figure 4 and Appendix A). In contrast, GLUT-1 overexpression in the cytoplasm and basement membrane of cancer cells from IPMN with INV was detected (Figure 4). Among the 15 patients with HGD IPMNs, the incidence of GLUT-1-positive expression in the cell membrane was 40 % (6/15) (Appendix A). GLUT-1 expression and the SUV-max corresponding to those of patients with IPMNs are summarized in Appendix A, and more than half of patients with HGD IPMNs had lower GLUT-1 expression.

### 3.4. Relationship between Mural Nodules and Preoperative Imaging Modality including FDG-PET/CT

Pathologically proven mural nodules were revealed in 68 patients (80.9%). Among them, PET uptake was positive in 44. When the pathologically proven mural nodules and FDG uptake were confirmed one-on-one by two clinicians (Y.H. and O.H.), out of the 44 patients in whom FDG uptake was observed, FDG was accumulated in 41 of their mural nodules (93%), excluding 3 in whom FDG was accumulated on the septum and cyst wall.

Additionally, there was a significant correlation between the size of the enhancing mural nodules measured by enhanced CT or enhanced EUS, and the pathological mural nodules (R = 0.7943, *p* < 0.001). Of the 33 patients with enhancing mural nodule height <5 mm and negative FDG uptake, only 1 (3.0%) INV was observed. Of the 23 patients with enhancing mural nodule height ≥10 mm and positive FDG uptake, 17 (73.9%) INVs were observed.

Among FDG-uptake-positive IPMNs, weak significant correlations were shown between mural nodule height and SUV-max (R = 0.1474, *p* = 0.0463) in all cases (Figure 5). However, when analyzed individually, there was no significant correlation among LGDs (R = 0.2349, *p* = 0.6190), HGDs (R = 0.1113, *p* = 0.3074), or INVs (R = −0.1901, *p* = 0.8976).

## 4. Discussion

As the definition of malignancy in IPMNs varies by country, region, and researcher, it has not yet been determined [12]. Most previous studies have included HGD and INV in one malignant category. Only a few studies have accurately assessed risk factors for each degree of dysplasia [27,28,29,30]. Thus, we focused on the diagnostic ability for the grade of dysplasia in IPMN using FDG-PET/CT. Based on what was revealed in the current study, we discuss the following three points in IPMNs: (1) differences in risk factors between HGD and INV; (2) FDG uptake and GLUT-1 expression in HGD and INV; and (3) the relationship between mural nodules and FDG in IPMNs.

First, treatment regimens for IPMNs involve surveillance protocol for LGD, minimally invasive surgery for LGD/HGD, and resection with lymph node dissection for INV. To facilitate surgical decision making for IPMNs, a highly accurate modality or finding that can distinguish the degree of malignancy in IPMN is required. Unfortunately, in our study, there were no useful factors that significantly separated the LGD, HGD, and INV groups from one another. However, enhancing mural nodule height and SUV-max were significantly higher in the INV group than in the LGD or HGD groups. Izumo et al. reported that the presence of an enhancing mural nodule height of ≥5 mm, concomitant pancreatitis, and wall and septal thickening were significant risk factors for HGD [8]. In contrast, MPD size of ≥10 mm, and an abrupt change in the caliber of the pancreatic duct with distal pancreatic atrophy, are risk factors for INV. However, contrary to the study by Izumo et al., our study showed that an MPD of ≥10 mm is a significant risk factor for HGD rather than INV. Moreover, enhancing mural nodules of ≥5 mm were also indicated as significant risk factors only for INV, but not for HGD. Multiple comparisons between the three groups (LGD, HGD, and INV) should be conducted, but Izumo et al. analyzed each of the two groups individually, which may have resulted in inaccurate statistical processing. These differences due to patient background may also have been affected [31]. From these conflicting results, no specific risk factors for HGD or INV could be determined. However, among the factors newly appearing during the follow-up of BD-IPMN, the appearance of mural nodules and MPD of ≥10 mm have been reported to be predictors of malignant IPMN [32]. Therefore, the appearance of any of these during the follow-up of BD-IPMN should warrant the consideration of the possibility of INV and surgical resection.

Second, regarding the diagnostic ability of FDG-PET/CT in IPMN, according to recent reviews, when the SUV-max cutoff was set between 1.3 and 3.0, FDG-PET/CT had malignant diagnostic ability, with a sensitivity of 80 (62–100)%, a specificity of 95 (71–100)%, and an accuracy of 87 (76–97)% [19,33]. However, these reviews include both studies on INV alone and studies on INV and HGD. In a recent study involving 171 patients with IPMN [22], SUV-max was significantly higher in HGD and INV than in LGD IPMN, and the sensitivity and specificity of SUV-max to distinguish benign (LGD) from malignant (INV and HGD) disease were 87.2% and 80.0%, respectively, when the cutoff value of SUV-max was set at 2.5. Moreover, several other reports analyzed the diagnostic ability of FDG-PET among LGD, HGD, and INV IPMNs; however, each report showed different results, and no significant conclusions could be obtained from the literature [23,24,25]. FDG is a glucose analog, and is moved into cells by glucose transporters, where it is then phosphorylated by hexokinases to FDG-6-phosphate [34]. In general, glucose-6-phosphatase activity in tumor cells is significantly reduced, and glucose transporters, which are involved in the intracellular transport of glucose, are often overexpressed. FDG-6-phosphate cannot be metabolized further in the glycolytic pathway, and stays in the cells [35]. Therefore, FDG uptake reflects glucose metabolism in the tissue, and activated cells enhance the uptake of FDG in many types of tumor cells relative to other normal cells [34]. Overexpression of GLUT-1 was reported to correlate with FDG uptake in various types of carcinoma (e.g., esophageal squamous-cell carcinoma, pancreatic carcinoma, and non-small-cell lung carcinoma) [26,35,36]. In the present study, the expression of GLUT-1 on immunohistochemical staining was almost nonexistent or weak among the HGD group, in which FDG uptake was low, whereas it was strong for the INV group, with its high FDG uptake. Hirashita et al. evaluated GLUT-1 and FDG-PET/CT using 39 IPMN pancreatic resection specimens, and found that the expression of GLUT-1 was significantly higher in carcinoma than in adenoma, showing that there is a correlation between SUV-max and the expression of GLUT-1 [37]. Oda et al. reported that HGD and INV showed higher expression of GLUT-1 than LGD [38]; their studies also found that there are tissue subtypes of IPMN, including many oncocytic types (*n* = 7) and the pancreatobiliary type (*n* = 21), which have higher expression of GLUT-1 than those seen in the gastric type (*n* = 30) and intensive type (*n* = 22). Our study did not include the oncocytic type, and the fact that only a few IPMNs were the pancreatobiliary type might have affected the difference in GLUT-1 expression for HGD. These results may indicate that HGD has oncologically low FDG uptake, and that FDG-PET/CT is unsuitable for HGD detection, although it is useful for differentiating between HGD and INV.

Third, how FDG accumulates at the site is unclear, as is its relationship with histopathological findings. Therefore, we investigated the relationship between FDG uptake and mural nodules, which are presumed to contain the most tumor cells. The height of enhancing mural nodules—presumed to contain tumor cells—has been considered a strong predictor of malignancy; however, even so, the PPV of HGD and INV is ~60%, and a malignant predictive diagnosis based on mural nodule height alone is not recommended [12,39,40,41,42]. In general, the problem is that SUVs have a partial volume effect when the lesion is small; thus, they may be underestimated [43]. In this study, IPMNs with HGD had a significantly lower mural nodule height than IPMNs with INV, which might be one of the reasons that FDG could not be detected. Previously, there was also no correlation found between mural nodules and FDG uptake [44]. Kawada et al. reported that they analyzed IPMNs with HGD or INV with a mural nodule of size 10 mm or more, finding that 23 out of 33 IPMNs (69.6%) had focal lesions of HGD or INV in mural nodules or outside the nodule. These results may indicate that the size of the mural nodules is not correlated with the volume of tumor cells with the highest degree of dysplasia [45]. Since the tumor volume of HGD or INV is not directly reflected in the mural nodule height, this may be one of the reasons that the association between FDG uptake and mural nodule height is absent. FDG-PET/CT is presumed to reflect the tumor volume of neoplastic cells with active glucose metabolism; thus, FDG-PET/CT may have additional value that mural nodule height does not have. It can be suggested that if the mural nodule height is 5 mm or less and the SUV-max is 4 or less, it is highly unlikely that INV tumor cells are contained; conversely, if the SUV-max is 4 or more and the mural nodule height is 10 mm or more, there is a high possibility that the lesion is HGD or INV. However, it should be noted that SUV-max has a higher cutoff value in this study than that reported in other studies. Since SUV values are affected by the PET-scanner, image viewer, and the method of drawing the volume of interest, these factors might have affected our current results [19,20,21,23].

Our study has several limitations. First, SUV-max can also be affected by factors such as tumor size or patient height and weight. Consequently, the higher malignant cutoff value than that in the previous reports may be due to differences in these factors. Second, in the current study, FDG uptake of IPMNs was evaluated using contrast-enhanced CT, MRI, and EUS findings as a guide to determine the portion with SUV-max, because it was quite difficult to detect the mural nodule by PET/CT alone. Third, the number of IPMNs with INV and HGD was small, while that of IPMNs with LGD was high. Thus, there may be a risk of generalization in the results. However, the ratio of LGDs was 51.2% of the study population, which is comparable to that reported in previous studies (32.2%–61.4%) [22,25,46,47]. In addition, there was no clear difference in SUV-max between LGD and HGD patients (*p* = 0.4921), and it was determined that a significant difference would not be obtained even if a larger research population was studied. Fourth, our study included patients who underwent pancreatectomy for IPMN, and had a selection bias because of the retrospective design. There were more patients in clinical practice without findings of suspicion of malignancy who were surveyed. Notably, data are only for patients with a high rate of malignancy in the population, and other risk factors for HGD or INV may not have been detected. Large-scale multicenter studies will reveal useful factors for differentiating LGD, HGD, and INV. The disadvantages of FDG-PET/CT are radiation exposure and high cost, and there was no clear difference in malignancy predictive ability compared to mural nodule height by CT, MRI, or EUS. However, FDG-PET/CT could determine whether the lesion might be invasive or non-invasive. IPMNs with INV may require more extensive pancreatectomy than non-invasive lesions and, therefore, our results offer critical information that may determine surgical treatment strategies. It may be necessary to elucidate new molecular oncological changes—which comprise differentially expressed genes and gene mutations that appear in the progression from LGD to HGD—and search for biomarkers.

## 5. Conclusions

FDG-PET/CT is useful in distinguishing between non-invasive and invasive IPMNs. Our results offer critical information that may determine surgical treatment strategies.

## Figures and Tables

**Figure 1 cancers-13-04633-f001:**
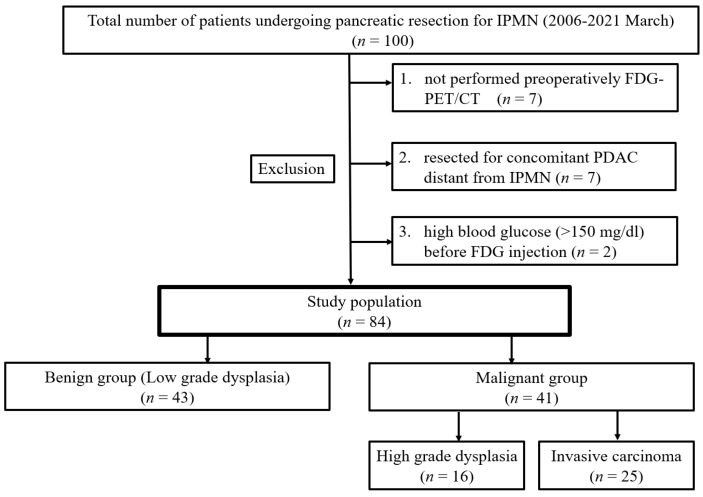
Distribution of the patients with IPMN. IPMN: intraductal papillary mucinous neoplasm; FDG: ^18^F-fluorodeoxyglucose.

**Figure 2 cancers-13-04633-f002:**
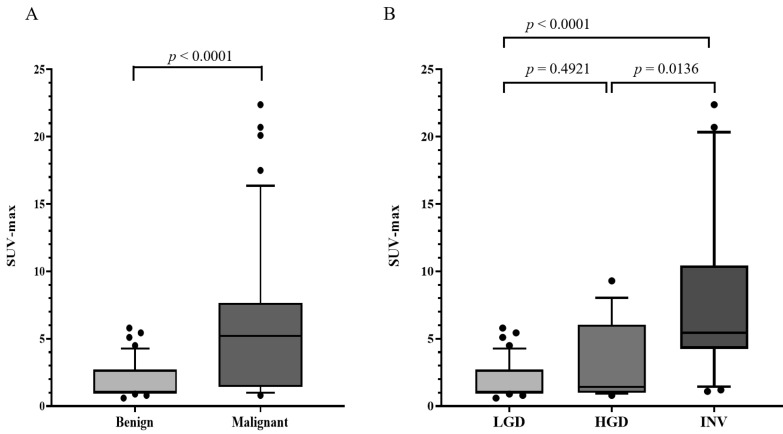
Box plot of the levels of SUV-max in the late phase. (**A**) Box plot of the levels of SUV-max in the benign group and malignant group are shown; SUV-max was significantly higher in the malignant group compared with the benign group (*p* < 0.0001). (**B**) Box plot of the levels of SUV-max in the LGD, HGD, and INV groups; SUV-max was significantly higher in the INV group that in the LGD and HGD groups (*p* < 0.0001, *p* = 0.0136, respectively). There was no significant difference in the levels of SUV-max between the LGD and HGD groups. SUV-max: maximum standardized uptake value; LGD: low-grade dysplasia; HGD: high-grade dysplasia; INV: invasive cancer.

**Figure 3 cancers-13-04633-f003:**
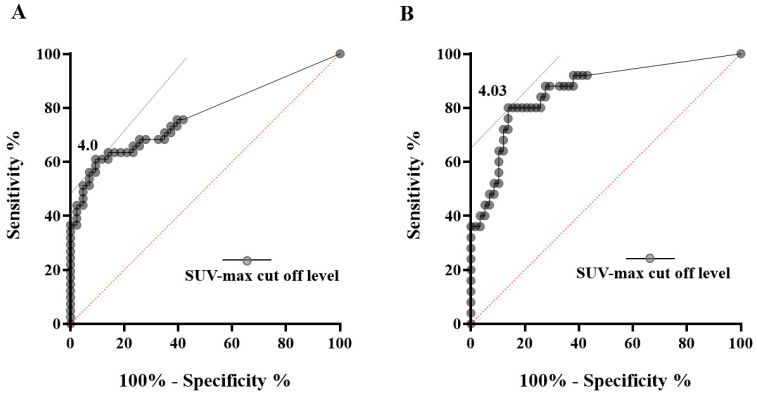
ROC curves. (**A**) Data on the curve represent SUV-max cutoff levels of 4.0 for the differential diagnosis of benign and malignant IPMNs according to the SUV-max of FDG-PET. (**B**) Data on the curve represent SUV-max cutoff levels of 4.03 for the differential diagnosis of low-grade dysplasia/high-grade dysplasia and invasive carcinoma according to the SUV-max of FDG-PET. ROC: receiver operating characteristic. SUV-max: maximum standardized uptake value; FDG-PET: 2-^18^F fluoro-2-deoxy-D-glucose positron emission tomography.

**Figure 4 cancers-13-04633-f004:**
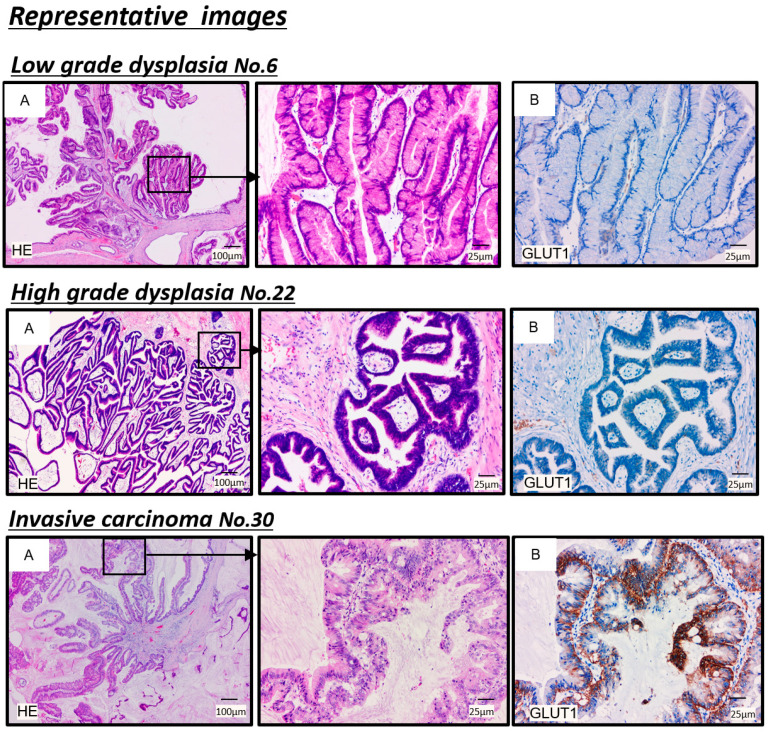
Representative hematoxylin and eosin (H&E)-stained images and immunohistochemical images for GLUT1 in IPMNs with low-grade dysplasia, high-grade dysplasia, and invasive carcinoma are shown. (**A**) H&E-stained sections of clinical specimens of IPMN. Areas in the boxed regions on the left are shown magnified on the right. (**B**) Immunohistochemical staining for GLUT-1. GLUT-1: glucose transporter 1; IPMN: intraductal papillary mucinous neoplasm.

**Figure 5 cancers-13-04633-f005:**
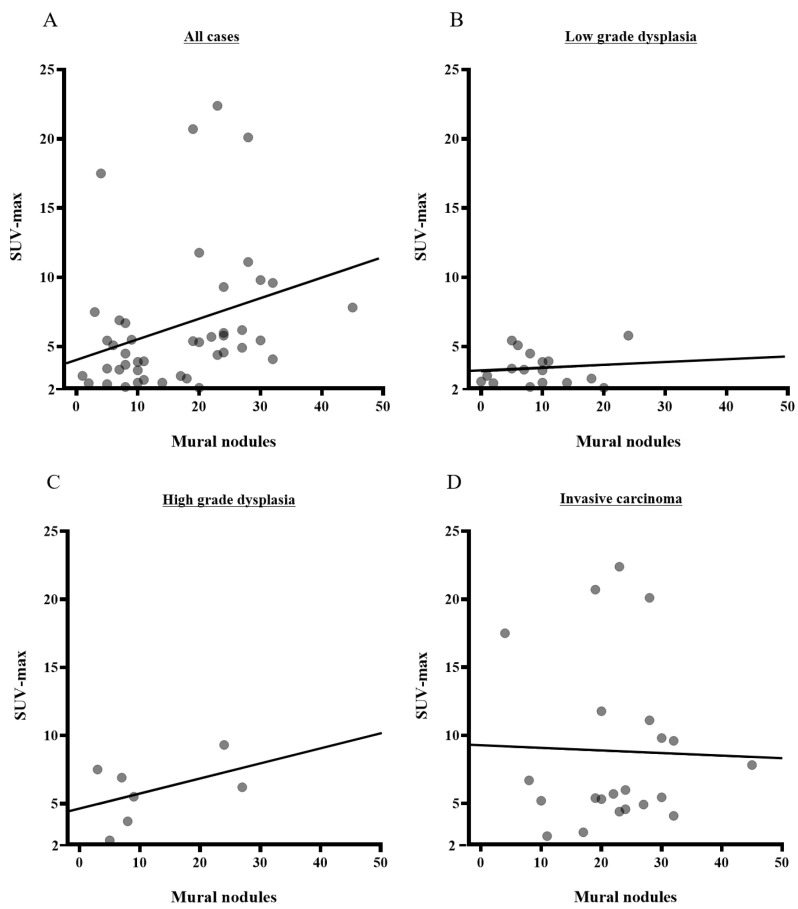
Correlations of the levels of SUV-max and mural nodule height by pathological grouping. (**A**) Significant correlations were found between mural nodule height and SUV-max in all IPMNs (R = 0.1474, *p* = 0.0463). (**B**) It was found that there was no significant correlation among LGDs (R = 0.2349, *p* = 0.6190). (**C**) It was found that there was no significant correlation among HGDs (R = 0.1113, *p* = 0.3074). (**D**) It was found that there was no significant correlation among INVs (R = −0.1901, *p* = 0.8976). SUV-max: maximum standardized uptake value; IPMN: intraductal papillary mucinous neoplasm.

**Table 1 cancers-13-04633-t001:** Patient background characteristics (*n* = 84).

Characteristic	Value
Age, y, median (range)	70 (35–87)
Sex, male, *n* (%)	56 (66.6)
Symptoms, presence, *n* (%)	28 (33.3)
History of pancreatitis, presence, *n* (%)	8 (9.5)
Family history of pancreatic cancer, *n* (%)	8 (9.5)
Coexistence of diabetes mellitus	23 (27.4)
Morphological subtype, *n* (%)	
Main duct type	9 (10.7)
Branch type	33 (39.3)
Mixed type	42 (50.0)
Location of IPMN, *n* (%)	
Head (including uncus)	49 (58.3)
Distal (left from SMV)	23 (27.4)
Multifocal	12 (14.3)
Histopathological diagnosis, *n* (%)	
Low grade dysplasia	43 (51.2)
High grade dysplasia	16 (19.0)
Invasive carcinoma	25 (29.8)

IPMN: Intraductal papillary mucinous neoplasm; SMV: superior mesenteric vein.

**Table 2 cancers-13-04633-t002:** Clinicopathological features between the benign and malignant groups.

Characteristic	Benign * (*n* = 43)	Malignant ^†^ (*n* = 41)	*p*-Value
Clinical factor			
Age, year, median (range)	70 (44–80)	70 (35–87)	0.4439
Sex, male, *n* (%)	34 (79.0)	22 (53.6)	0.0128
Symptoms, presence, *n* (%)	7 (16.2)	21 (51.2)	0.0006
Coexistence of diabetes mellitus, *n* (%)	8 (18.6)	15 (36.6)	0.0633
Concomitant pancreatitis, *n* (%)	4 (9.3)	4 (9.8)	0.9449
Family history of pancreatic cancer, *n* (%)	4 (9.3)	3 (7.3)	0.7416
Laboratory factor			
Serum CEA, high (≥ 3.2 IU/L)	10 (23.2)	18 (43.9)	0.0438
Serum CA 19-9, high (≥ 37 IU/L)	2 (4.6)	10 (24.3)	0.0074
Serum P-AMY, high (≥ 50 IU/L)	11 (25.5)	8 (19.5)	0.5055
Imaging factor			
Morphological subtype, (MD/BD/mixed), *n*	1/16/24	8/15/18	0.0559
Main pancreatic duct size, mean ± SD	6.51 ± 3.90	8.63 ± 5.74	0.0484
Main pancreatic duct size, (≥ 10 mm), *n* (%)	7 (16.3)	17 (41.4)	0.0210
Cystic size, mean ± SD	34.7 ± 14.1	39.8 ± 23.6	0.2194
Enhancing mural nodule height ^‡^, mean ± SD	3.67 ± 6.01	14.6 ± 12.4	<0.0001
Enhancing mural nodule height ^‡^, (≥ 5 mm), *n* (%)	12 (27.9)	29 (70.7)	<0.0001
Cyst growth rate ≥ 5 mm/2 years, *n* (%)	3 (7.0)	2 (4.9)	0.6833
Abrupt change in caliber of pancreatic duct with distal pancreatic atrophy, *n* (%)	1 (2.3)	1 (2.4)	0.9728
Thickened/enhancing cyst walls, *n* (%)	12 (27.9)	2 (4.9)	0.0164
Lymphadenopathy, *n* (%)	1 (2.3)	1 (2.3)	0.9728
FDG uptake, positive, *n* (%)	16 (37.2)	28 (68.2)	0.0040
SUV-max, mean ± SD	1.95 ± 1.39	5.96 ± 5.60	<0.0001
Other indicators			
High-risk stigmata, positive, *n* (%)	17 (39.5)	34 (82.9)	<0.0001
Worrisome feature, positive, *n* (%)	41 (95.3)	40 (97.6)	0.5811

* Benign indicates low-grade dysplasia. ^†^ Malignant indicates high-grade dysplasia or invasive carcinoma. ^‡^ The mural nodule height was measured by contrast-enhanced computed tomography. BD: branch duct; CA19-9: carbohydrate antigen19-9; CEA: carcinoembryonic antigen; FDG: fluorodeoxyglucose; MD: main duct; P-AMY: pancreatic amylase; SD: standard deviation; SUV: standardized uptake value.

**Table 3 cancers-13-04633-t003:** Clinicopathological features between low-grade dysplasia, high-grade dysplasia, and invasive carcinoma.

Characteristic	LGD (*n* = 43)	HGD (*n* = 16)	INV (*n* = 25)	*p*-Value (LGD vs. HGD)	*p*-Value (LGD vs. INV)	*p*-Value (HGD vs. INV)
Clinical factor						
Age, median (range), y	70 (44–80)	70 (49–86)	70 (35–87)	0.5400	0.9126	0.7659
Sex, male, *n* (%)	34 (79.0)	9 (56.3)	13 (52.0)	0.1951	0.0552	0.9668
Symptoms, presence, *n* (%)	7 (16.2)	7 (43.8)	14 (56.0)	0.0755	0.0021	0.7393
Coexistence of diabetes mellitus	8 (18.6)	7 (43.8)	8 (32.0)	0.1265	0.4309	0.7417
Concomitant pancreatitis, presence, *n* (%)	4 (9.3)	2 (12.5)	2 (8.0)	0.5933	0.2701	0.0726
Family history of pancreatic cancer	4 (9.3)	1 (6.3)	2 (9.1)	0.9335	0.9845	0.9827
Laboratory findings						
Serum CEA, high (≥3.2 IU/L)	10 (23.2)	4 (25.0)	14 (56.0)	0.8737	0.0709	0.0928
Serum CA 19-9, high (≥37 IU/L)	2 (4.6)	1 (6.3)	9 (36.0)	0.7282	0.0043	0.0957
Serum P-AMY, high (≥50 IU/L)	11 (25.5)	2 (12.5)	6 (24.0)	0.1904	0.0495	0.6175
Imaging findings						
Main pancreatic duct size, mean ± SD	6.51 ± 3.90	9.05 ± 4.93	8.36 ± 6.19	0.0900	0.7057	0.7815
Main pancreatic duct size, (≥ 10 mm), *n* (%)	7 (16.3)	8 (50.0)	10 (40.0)	0.0456	0.2556	0.6649
Cystic size, mean ± SD	34.7 ± 14.1	38.1 ± 28.9	40.9 ± 19.3	0.9885	0.5333	0.7344
Enhancing mural nodule height *, mean ± SD	3.67 ± 6.01	6.56 ± 8.36	19.7 ± 11.9	0.3411	< 0.0001	0.0045
Enhancing mural nodule height *, (≥ 5 mm), *n* (%)	12 (27.9)	8 (50.0)	29 (84.0)	0.2588	< 0.0001	0.0574
Cyst growth rate ≥ 5 mm/2 years, presence, *n* (%)	3 (7.0)	0 (0)	2 (8.0)	0.5435	0.9892	0.5079
Abrupt change in caliber of pancreatic duct with distal pancreatic atrophy	1 (2.3)	1 (6.3)	0 (0)	0.7593	0.7445	0.4531
Thickened/enhancing cyst walls, *n* (%)	12 (27.9)	1 (6.3)	1 (4.0)	0.1842	0.0444	0.9558
Lymphadenopathy, *n* (%)	1 (2.3)	1 (6.3)	0 (0)	0.7593	0.7445	0.4531
SUV-max, mean ± SD	1.95 ± 1.39	3.17 ± 2.82	7.91 ± 6.06	0.4921	< 0.0001	0.0136
Other indicators						
High-risk stigmata, positive, *n* (%)	17 (39.5)	12 (75.0)	22 (88.0)	0.0442	0.0003	0.5482
Worrisome feature, positive, *n* (%)	41 (95.3)	16 (100.0)	24 (96.0)	0.6762	0.9937	0.7336

CA19-9: carbohydrate antigen19-9; CEA: carcinoembryonic antigen; HGD: high-grade dysplasia; LGD: low-grade dysplasia; INV: invasive carcinoma; P-AMY: pancreatic amylase; SD: standard deviation; SUV: standardized uptake value. * The mural nodule height was measured by contrast-enhanced computed tomography.

## Data Availability

The data presented in this study are available on request from the corresponding author.

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
