# Peer review of "Clinical Utility and Limitation of Diagnostic Ability for Different Degrees of Dysplasia of Intraductal Papillary Mucinous Neoplasms of the Pancreas Using 18F-Fluorodeoxyglucose-Positron Emission Tomography/Computed Tomography"

_cancers, 2021, doi:10.3390/cancers13184633_

Round 1
Reviewer 1 Report
This study aimed to assess the role of FDG uptake in the diagnosis of high-grade dysplasia (HGD) of IPMN. It was concluded that FDG-positron-emission tomography/computed tomography is useful to predict INV, while not HGD.
However, as the author stated, there is too much limitation in this paper. It is questionable whether the results of this study will have any usefulness in the actual clinical practice of IPMN.
1
As described by the author, there have been many reports on the usefulness of PET/CT for the diagnosis of tumor malignancy. However, one of the characteristics of PET/CT is that the size of the tumor affects the findings. In addition, cystic tumors are more difficult to visualize than mass lesions such as GISTs and pancreatic tumors. This paper concludes that PET/CT is useful in differentiating benign from malignant IPMNs (especially in differentiating LGD/HGD from INV), but how is an IPMN with a nodule height of 5 mm depicted by PET/CT? How is an IPMN with a nodal height of 5 mm visualized by PET/CT? What part of the nodule was used to measure SUV-max?
MRCP/EUS/PET-CT should be presented as Figure for LGD, HGD, INV, respectively.
2
In the background comparison of benign and malignant IPMNs, the author should explain that the nodule height is higher in benign than in malignant. (Benign 14.6 ± 12.4 vs Malignant 3.67 ± 6.01)
With which modality was the nodule height measured?
Were there any cases where it was a mucous mass rather than a nodule?
Author Response
Manuscript ID: cancers-1346147
Title: Clinical utility and limitation of diagnostic ability for different degrees of dysplasia of intraductal papillary mucinous neoplasms of the pancreas using 18F-fluorodeoxyglucose-positron emission tomography/computed tomography
Response to Reviewer 1 Comments:
Comment-#1:
This study aimed to assess the role of FDG uptake in the diagnosis of high-grade dysplasia (HGD) of IPMN. It was concluded that FDG-positron-emission tomography/computed tomography is useful to predict INV, while not HGD. However, as the author stated, there is too much limitation in this paper. It is questionable whether the results of this study will have any usefulness in the actual clinical practice of IPMN. As described by the author, there have been many reports on the usefulness of PET/CT for the diagnosis of tumor malignancy. However, one of the characteristics of PET/CT is that the size of the tumor affects the findings. In addition, cystic tumors are more difficult to visualize than mass lesions such as GISTs and pancreatic tumors. This paper concludes that PET/CT is useful in differentiating benign from malignant IPMNs (especially in differentiating LGD/HGD from INV), but how is an IPMN with a nodule height of 5 mm depicted by PET/CT? How is an IPMN with a nodal height of 5 mm visualized by PET/CT? What part of the nodule was used to measure SUV-max? MRCP/EUS/PET-CT should be presented as Figure for LGD, HGD, INV, respectively.
Response-#1: We deeply appreciate your constructive comments. We obtained valuable results that FDG-PET/CT could determine whether the lesion might be invasive or non-invasive. The distinction between invasive and non-invasive IPMN is important when deciding the surgical treatment approach, but there have been few reports focusing on the diagnostic ability of SUV-max to discriminate HGD from INV. Thus, our results may offer critical information to determine surgical treatment strategies in patients planned for undergo resection. In addition, our paper clarified the relationship between SUV-max and mural nodules, and therefore, we believe that this research is meaningful. Regarding tumor size, IPMN with HGD has a significantly smaller mural nodule height than INV, which might be one of the reasons why FDG could not be sufficiently taken up in HGD. As you point out, it is difficult to depict small mural nodules by PET/CT alone; thus, we evaluated the IPMN area with maximum FDG uptake using contrast-enhanced CT, MRI, and EUS findings as a guide. We calculated the SUV-max for the IPMN region with the highest FDG uptake, including mural nodule, cyst wall, or septa. We have added the following text in response to your comment and added the representative images of each modality for LGD, HGD, and INV in Figure S1.
Page 1: Abstract
“The diagnostic value of 18F-fluorodeoxyglucose (FDG) uptake in the management of intraductal papillary mucinous neoplasm (IPMN) of the pancreas remains unclear. This study aimed to assess the role of FDG uptake in the diagnosis of different degrees of dysplasia of IPMN. We retrospectively analyzed the following three points in 84 patients with IPMN: (1) risk factors to predict high-grade dysplasia (HGD) and invasive carcinoma (INV), (2) the relationship between FDG uptake and glucose transporter 1 (GLUT-1) expression, and (3) the relationship between FDG uptake and the presence of mural nodules. The histopathological diagnosis was low-grade dysplasia (LGD) in 43 patients, HGD in 16, and INV in 25. The maximum standardized uptake value (SUV-max) was significantly higher in INV than in LGD/HGD (P < .0001, P = .0136). The sensitivity and specificity to discriminate INV from LGD/HGD were 80.0% and 86.2%, respectively, using the receiver operator characteristic curve, when optimal cut-off score of SUV-max was set at 4.03. Those values were not different between HGD and LGD. More than half of HGD patients had low GLUT-1 expression. Taken together, FDG-PET/CT is useful in distinguishing between non-invasive and invasive IPMN. Our results offer critical information that may determine surgical treatment strategies.”
Page 2, lines 70-73:
“Additionally, we occasionally need more information about the degree of dysplasia of IPMN depending on the patient’s age, general condition, tumor location, etc. In particular, during surgical decision making, it is important to determine whether the lesion is non-invasive or invasive IPMN.”
Page 3, lines 113-115:
“We performed contrast-enhanced CT, followed by MRI and EUS in all patients during the initial assessment of IPMNs. Then, PET/CT was conducted in patients who were planned to undergo surgical resection.”
Page 3, lines 118-120:
“We evaluated the IPMN area with maximum FDG-uptake using contrast-enhanced CT, MRI, and EUS findings as a guide.”
Page 3, lines 121-124:
“SUV-max equals the organic radioactivity ((MB q/g)/18F-FDG (MBq/g body weight)) and was calculated in the IPMN region with the highest FDG uptake. For example, the value was maximum in the mural nodule in some IPMNs, but in the cyst wall or septa in others.”
Page 12, lines 358-361:
“In general, the problem is that SUVs have a partial volume effect when the lesion is small; thus, they may be underestimated [43]. In this study, IPMN with HGD had a significantly lower mural nodule height than IPMV with INV, which might be one of the reasons why FDG could not be detected. ”
Page 12, lines 381-384:
“Second, in the current study, FDG uptake of IPMN was evaluated using contrast-enhanced CT, MRI, and EUS findings as a guide to determine the portion with SUV-max because it was quite difficult to detect the mural nodule by PET/CT only.”
Page 13, lines 398-402:
“However, FDG-PET/CT could determine whether the lesion might be invasive or non-invasive. IPMN with INV may require more extensive pancreatectomy than non-invasive lesion, and therefore, our results offer critical information that may determine surgical treatment strategies.”
Page 13, lines 405-407:
“Conclusions
FDG-PET/CT is useful in distinguishing between non-invasive and invasive IPMN. Our results offer critical information to determine surgical treatment strategies.”
Comment-#2:
In the background comparison of benign and malignant IPMNs, the author should explain that the nodule height is higher in benign than in malignant. (Benign 14.6 ± 12.4 vs Malignant 3.67 ± 6.01)
Response-#2: Thank you for pointing out this issue. We deeply apologize for the careless mistake in the values of the mural nodule. In fact, the value of malignant was 14.6 ± 12.4, while that of benign was 3.67 ± 6.01. We have corrected these values in the revised version of Table. 1.
Comment-#3:
With which modality was the nodule height measured?
Response-#3: We apologize for not providing sufficient information. In this study, the mural nodule height was determined using contrast-enhanced CT imaging. We have added the following text regarding the measurement method of mural nodule height.
Page 3, line 128- Page 4, line129:
“ 2.4. Evaluation of mural nodule height
We measured mural nodule height using contrast-enhanced CT.”
Comment-#4:
Were there any cases where it was a mucous mass rather than a nodule?
Response-#4: Thank you for pointing out this critical issue. In this study, no patient had mucous mass that was initially judged as mural nodule. We have added the following text in the “Results” part.
Page 4, lines 172-173:
“No patient had mucous mass that was initially judged as mural nodule.”

Reviewer 2 Report
Hozaka et al. showed the diagnostic ability of FDG-PET/CT for IPMN, and they concluded that FDG-PET/CT is useful in distinguishing between low-grade dysplasia, high-grade dysplasia, and invasive carcinoma.
Authors need to be commended on their efforts to provide a great impact for screening diagnosis for IPMN. I think the value of this manuscript as an original article seems to be suitable for publication in the journal.
Author Response
Manuscript ID: cancers-1346147
Title:Clinical utility and limitation of diagnostic ability for different degrees of dysplasia of intraductal papillary mucinous neoplasms of the pancreas using 18F-fluorodeoxyglucose-positron emission tomography/computed tomography
Response to Reviewer 2 Comments:
Comment-#1:
Hozaka et al. showed the diagnostic ability of FDG-PET/CT for IPMN, and they concluded that FDG-PET/CT is useful in distinguishing between low-grade dysplasia, high-grade dysplasia, and invasive carcinoma.
Authors need to be commended on their efforts to provide a great impact for screening diagnosis for IPMN. I think the value of this manuscript as an original article seems to be suitable for publication in the journal.
Response-#1: We deeply appreciate your comment. This paper was submitted for English proofreading according to the reviewers` suggestion.

Reviewer 3 Report
The authors analyzed 84 IPMN patients who underwent surgical resection following PET scan. Despite the limitations of the retrospective study, the study was well designed.
Since the number of patients with invasive IPMN and High-grade dysplasia was too small, there is a risk of generalization error. Also, the sensitivity and specificity should be validated in other cohorts.
Author Response
Manuscript ID: cancers-1346147
Title:Clinical utility and limitation of diagnostic ability for different degrees of dysplasia of intraductal papillary mucinous neoplasms of the pancreas using 18F-fluorodeoxyglucose-positron emission tomography/computed tomography
Response to Reviewer 3 Comments:
Comment-#1:
The authors analyzed 84 IPMN patients who underwent surgical resection following PET scan. Despite the limitations of the retrospective study, the study was well designed. Since the number of patients with invasive IPMN and High-grade dysplasia was too small, there is a risk of generalization error. Also, the sensitivity and specificity should be validated in other cohorts.
Response-#1: We appreciate your valuable comment. As you have pointed out, the possibility of generalization error cannot be denied. Several other reports have analyzed the diagnostic ability of FDG-PET among LGD, HGD, and INV, but no large cohort has been used to validate the malignancy diagnostic ability of PET/CT in IPMNs. Only one report provided the results based on a larger cohort than ours. We have cited this paper and added the following text as one of the limitations of this study.
Page 12, lines 384-390:
“Third, the number of IPMNs with INV and HGD was small, while that of IPMNs with LGD was high. Thus, there may be a generalization risk in the results. However, the ratio of LGDs was 51.2% of the study population, which was comparable to that reported in previous studies (32.2%–61.4%) [22,25,46,47]. In addition, there was no clear difference in the SUV-max between LGD and HGD patients (p=.4921), and it was determined that a significant difference would not be obtained even if a larger research population was studied.”
Page 11, lines 321-327:
“In a recent study involving 171 patients with IPMN [22], SUV-max was significantly higher in HGD and INV than in LGD IPMN, and the sensitivity and specificity of SUV-max to distinguish benign (LGD) from malignant (INV and HGD) disease was 87.2% and 80.0%, respectively, when the cut-off value of the SUV-max was set at 2.5. Also, several other reports analyzed the diagnostic ability of FDG-PET among LGD, HGD, and INV IPMNs; however, each report showed different results, and no significant conclusion could be obtained in the literature [23-25].”

Reviewer 4 Report
It is an in-depth study with PET information which is important pre-operative examinations, clinical information, and IHC analysis.
However, its actual clinical usefulness in study design is questionable.
The most important consideration before treatment for Pancreas IPMN is the patient's choice of surgical resection.
The most important differentiation focus is to differentiate between LGD and HGD.
However, according to the results of this paper, it is not possible to differentiate between LGD and HGD with PET.
It is not helpful at all in determining the most important treatment policy.
And the PET SUV max mentioned in this paper is not a new novel marker.
In fact, it is a widely used clinical marker, but the number of cases is small to draw conclusions from this.
Moreover, the primary goal is to identify HGD, but the number of HGD is only 16.
It was said that FG was followed as the criteria for determining the surgical indication, and 43 out of 84 patients were LGD.
This is probably because the registration period is long, so the indication has changed.
It is recommended to provide specific reasons for resection.
During surveillance of IPMN, a borderline disease,
It is thought that the usefulness of PET as a primary test is low.
Rather, it is thought that CT and MRI will be sufficient for resection selection.
To support this result, comparison with conventional MRI and EUS diagnostic results is required.
I think the analysis that looked at the relationship with the mural nodule size is a good analysis.
Author Response
Manuscript ID: cancers-1346147
Title:Clinical utility and limitation of diagnostic ability for different degrees of dysplasia of intraductal papillary mucinous neoplasms of the pancreas using 18F-fluorodeoxyglucose-positron emission tomography/computed tomography
Response to Reviewer 4 Comments:
Comment-#1:
It is an in-depth study with PET information which is important pre-operative examinations, clinical information, and IHC analysis. However, its actual clinical usefulness in study design is questionable. The most important consideration before treatment for Pancreas IPMN is the patient's choice of surgical resection. The most important differentiation focus is to differentiate between LGD and HGD. However, according to the results of this paper, it is not possible to differentiate between LGD and HGD with PET. It is not helpful at all in determining the most important treatment policy.
Response-#1: We would like to thank you for your meaningful comments. We failed to show the role of FDG-PET/CT in discriminating HGD from LGD, which would be useful for critical decision-making during management of IPMN, whether resection or surveillance without resection. However, we fortunately obtained an unexpected result that FDG-PET/CT could determine whether the lesion is invasive or non-invasive in patients who are planned to undergo resection. Invasive IPMN would require more extensive pancreatectomy than non-invasive lesion, and therefore, our results offer critical information to determine surgical treatment strategies. We have added the following text in response to your comment.
Page 1: Abstract
“The diagnostic value of 18F-fluorodeoxyglucose (FDG) uptake in the management of intraductal papillary mucinous neoplasm (IPMN) of the pancreas remains unclear. This study aimed to assess the role of FDG uptake in the diagnosis of different degrees of dysplasia of IPMN. We retrospectively analyzed the following three points in 84 patients with IPMN: (1) risk factors to predict high-grade dysplasia (HGD) and invasive carcinoma (INV), (2) the relationship between FDG uptake and glucose transporter 1 (GLUT-1) expression, and (3) the relationship between FDG uptake and the presence of mural nodules. The histopathological diagnosis was low-grade dysplasia (LGD) in 43 patients, HGD in 16, and INV in 25. The maximum standardized uptake value (SUV-max) was significantly higher in INV than in LGD/HGD (P < .0001, P = .0136). The sensitivity and specificity to discriminate INV from LGD/HGD were 80.0% and 86.2%, respectively, using the receiver operator characteristic curve, when optimal cut-off score of SUV-max was set at 4.03. Those values were not different between HGD and LGD. More than half of HGD patients had low GLUT-1 expression. Taken together, FDG-PET/CT is useful in distinguishing between non-invasive and invasive IPMN. Our results offer critical information that may determine surgical treatment strategies.”
Page 2, lines 70-73:
“Additionally, we occasionally need more information about the degree of dysplasia of IPMN depending on the patient’s age, general condition, tumor location, etc. In particular, during surgical decision making, it is important to determine whether the lesion is non-invasive or invasive IPMN.”
Page 13, lines 399-402:
“FDG-PET/CT could determine whether the lesion might be invasive or non-invasive. IPMN with INV may require more extensive pancreatectomy than non-invasive lesion, and therefore, our results offer critical information that may determine surgical treatment strategies.”
Comment-#2:
And the PET SUV max mentioned in this paper is not a new novel marker. In fact, it is a widely used clinical marker, but the number of cases is too small to draw conclusions from this. Moreover, the primary goal is to identify HGD, but the number of HGD is only 16. It was said that FG was followed as the criteria for determining the surgical indication, and 43 out of 84 patients were LGD. This is probably because the registration period is long, so the indication has changed. It is recommended to provide specific reasons for resection.
Response-#2: Thank you for your valuable comment. The previous 3 GLs provided malignancy predictors, including high-risk stigmata or worrisome features; these are just malignancy predictors but not indicators of resection. Therefore, surgical indication varied by era. As you have pointed out, the number of IPMNs with INV and HGD was relatively small, and the ratio of LGD was high; thus, it cannot be denied that the proportion of the various grades of IPMNs in this study might have affected the results. However, the ratio of LGD in this study is comparable to that of previous studies (32.2%–61.4%). Therefore, we have added the following text to the limitations section.
Page 12, lines 384-390:
“Third, the number of IPMNs with INV and HGD was small, while that of IPMNs with LGD was high. Thus, there may be a generalization risk in the results. However, the ratio of LGDs was 51.2% of the study population, which was comparable to that reported in previous studies (32.2%–61.4%) [22,25,46,47]. In addition, there was no clear difference in the SUV-max between LGD and HGD patients (p=.4921), and it was determined that a significant difference would not be obtained even if a larger research population was studied.”
Comment-#3:
During surveillance of IPMN, a borderline disease, it is thought that the usefulness of PET as a primary test is low. Rather, it is thought that CT and MRI will be sufficient for resection selection. To support this result, comparison with conventional MRI and EUS diagnostic results is required. I think the analysis that looked at the relationship with the mural nodule size is a good analysis.
Response-#3: We deeply appreciate your advice. Regarding benign/malignant discrimination, AUC of SUV-max was 0.771. In contrast, when the ROC curve was constructed using mural nodule height measured by CT, MRI, and EUS, AUCs were 0.762, 0.731, and 0.785, respectively. As you have pointed out, the additional value of FDG-PET/CT was low; however, SUV-max provided the additional information in patients planned for resection. Taken together, we think that the findings of conventional imaging modalities are sufficient to determine whether the intended lesions should be resected or surveyed without resection. On the other hand, FDG-PET/CT could determine whether the lesion might be invasive or non-invasive in patients who planned to undergo resection. IPMN with INV would require more extensive pancreatectomy than non-invasive lesion, and therefore, our results offer critical information that may determine surgical treatment strategies. We believe that this research is meaningful, and have added the following text regarding the diagnostic ability of CT, MRI, and EUS. We have also changed the “Conclusion” part.
Page 9, lines 239-242:
“When the ROC curve was constructed using the mural nodule height measured by CT, MRI, and EUS for benign/malignant discrimination, optimal cut-off values and AUC were as follows: CT 4.8 mm (AUC 0.762), MRI 5.4 mm (AUC 0.731), and EUS 9.9 mm, (AUC 0.785).”
Page 13, lines 396-402:
“The disadvantage of FDG-PET/CT was radiation exposure and high cost, and there was no clear difference in malignancy predictive ability compared to mural nodule height by CT, MRI, or EUS. However, FDG-PET/CT could determine whether the lesion might be invasive or non-invasive. IPMN with INV may require more extensive pancreatectomy than non-invasive lesion, and therefore, our results offer critical information that may determine surgical treatment strategies.”
Page 13, lines 405-407:
“5. Conclusions
FDG-PET/CT is useful in distinguishing between non-invasive and invasive IPMN. Our results offer critical information that may determine surgical treatment strategies.”

Round 2
Reviewer 1 Report
The author has made sufficient text corrections to the Reviewer's suggestions.
The modified text is deemed useful to the reader.
Reviewer 4 Report
Good revision